# Modulation of the Morphological Architecture of Mn_2_O_3_ Nanoparticles to MnCoO Nanoflakes by Loading Co^3+^ Via a Co-Precipitation Approach for Mosquitocidal Development

**DOI:** 10.3390/mi14030567

**Published:** 2023-02-27

**Authors:** Rania A. Mohamed, Lamyaa M. Kassem, Niveen M. Ghazali, Elsayed Elgazzar, Wageha A. Mostafa

**Affiliations:** 1Department of Biology, Deanship of Educational Services, Qassim University, P.O. Box 5888, Unaizah 56219, Qassim, Saudi Arabia; 2Parasitology Department, Faculty of Veterinary Medicine, Zagazig University, P.O. Box 44519, Zagazig 44516, Egypt; 3Department of Pharmacy Practice, Unaizah College of Pharmacy, Qassim University, P.O. Box 5888, Unaizah 51911, Qassim, Saudi Arabia; 4Department of Pharmaceutical Chemistry and Pharmacognozy, Unaizah College of Pharmacy, Qassim University, P.O. Box 5888, Unaizah 51911, Qassim, Saudi Arabia; 5Department of Physics, Faculty of Science, Suez Canal University, Ismailia 41522, Egypt; 6Entomology Section, Zoology Department, Faculty of Science, Zagazig University, Zagazig 44519, Egypt

**Keywords:** Mn_2_O_3_ nanocubes, MnCoO nanoflakes, energy gap, *Culex pipiens*, larvicidal, pupicidal, midgut deformation

## Abstract

The spread of many infectious diseases by vectors is a globally severe issue. Climate change and the increase of vector resistance are the primary sources of rising mosquito populations. Therefore, advanced approaches are needed to prevent the dispersal of life-threatening diseases. Herein, Mn_2_O_3_ NPs and MnCoO nanocomposites were presented as mosquitocidal agents. The synthesized samples were prepared by a co-precipitation route and characterized using different techniques indicating the change of host Mn_2_O_3_ structure to 2D MnCoO nanoflakes with Co^3+^ integration. The thermal decomposition of the nanoparticles was examined by TGA analysis, showing high stability. The energy gap ($E_{g}$) of Mn_2_O_3_ was estimated within the visible spectrum of the value 2.95 eV, which reduced to 2.80 eV with doping support. The impact of Mn_2_O_3_ and MnCoO on immature stages was investigated by semithin photomicrographs exhibiting significant changes in the midgut, fat tissue and muscles of the third larval instar. Moreover, the external deformations in pupae were examined using scanning electron microscopy (SEM).

## 1. Introduction 

Technological developments in the field of industry and the rise of pollution percentages represent a major threat for human life. One of the essential drawbacks of global warming is the growing number of hazardous microbes and mosquitoes’ vectors which transmit varieties of infectious diseases that have deleterious public health impacts [1,2,3]. Mosquitoes have the leading role in transmitting fatal diseases such as malaria and filariasis, yellow fever, dengue fever, and zika viruses [4]. For example, *Culex pipiens* is the widespread household mosquito that works as a vector of various life-threatening diseases, including West Nile virus, encephalitis virus, rift valley fever, and filariasis by *Wuchereria bancrofti* in humans [5]. Although climate change is constantly defying the world’s populations, mosquitoes are receiving benefits from this competition. When the temperature rises, mosquitoes propagate faster and spread, carrying disease [6]. Due to the failure of traditional insecticides to eliminate infectious vectors, mosquitoes have developed elevated levels of resistance. Additionally, these chemical pesticides are harmful and have unfavorable consequences for many living organisms [7]. Recently, several studies have been performed universally for developing novel strategies to avoid this crucial issue. Nanotechnology is demonstrated as a promising solution for mosquito control [8,9]. Nanomaterials and their derivatives, including noble nanometals (i.e., Au, Ag, Ru), carbonaceous nanomaterials, CNTs, GO, graphene, as well as the nanometal oxides, have gained great attention in the present decade [10,11,12]. Nowadays, metal oxide semiconductors are used in electronics, solar cells, and photocatalysis. ZnO, CuO, MgO, and AgO nanoparticles (NPs) have approved excellent performance in biomedical contexts, environmental remediation, food security, and agriculture [13,14]. Manganese oxide (MnO) is an important compound owing to its outstanding features, including various oxidation states (MnO, Mn_2_O_3_, MnO_2_, and Mn_3_O_4_), large surface to volume area, and high porosity. Mn_2_O_3_ NPs have unique magnetic aspects, stable structure, and low toxicity, in addition to their environmental benefits, which have been applied in supercapacitors, electrochemical analysis, lithium-ion batteries, and biosensors [12,13,14]. M. Kelani et al. have reported the high sensitivity of Mn_2_O_3_/MCNTs for the detection of the dinitolmide drug ratio in various food samples [15]. Moreover, manganese oxide is regarded as an efficient chemical in medicine and healthcare. Jiansheng Lin et al. have investigated the biomedical development of porous manganese dioxide nanoparticles (MnO_2_ NPs) on the microstructure of red blood cells and functions [16]. Despite these advantages, manganese oxide has a few drawbacks, such as weak electrical behavior and wide band gap. Some past studies have reported the ability to amend manganese oxide characteristics by doping with different transition metals such as Fe, Cu, Ni, Ag, and Co, for tuning the optical band gap and morphological structure. Siddique et al. have improved the photocatalytic activity and gas sensing performance-based manganese oxide by zinc ions (Zn^2+^) replacement [17]. Moreover, architecture structures such as nanorods, nano comb, nanoflower, nanosheets, etc., are valuable for boosting the density of free charge carriers and surface defects (oxygen vacancies and zinc interstitial). In the present study, Mn_2_O_3_ nanocubes were incorporated with cobalt ions (Co^3+^) and the two-dimensional Mn_2_O_3_/Co_3_O_4_ (2D MnCoO) nanoflakes were obtained [18]. Generally, 2D NPs provide more free carriers attributed to quantum size effect and the topological architectures [19,20]. It is expected that the integration of Mn_2_O_3_ with Co^3+^ will generate more reactive oxygen species (ROS), which play a significant role in larvicidal activity and biological applications. In recent studies, MnCoO/CNT nanoflakes and Co_3_O_4_ nanorods were utilized as larvicidal agents for inhibiting the enzyme activities of mosquito *C. pipiens* larvae, resulting in larval death [12,21]. Numerous techniques are employed for fabricating the nanomaterials comprising chemical bath deposition, sol-gel, chemical vapor deposition, and electrodeposition methods. Among the chemical methods, co-precipitation is available, cheap, and has an easily controllable shape and particle size [21,22]. The main target of the current study is to assess novel eradication methods for restrain pathogenic vectors based on modifications to the structural-morphological merits of the Mn_2_O_3_ NPs.

## 2. Experimental Section

### 2.1. Reagents

Manganese chloride (MnCl_2_), cobalt chloride hexahydrate (CoCl_2_.6H_2_O), ethyl alcohol, 2.5% glutaraldehyde, 0.1 M cacodylate buffer, osmium tetroxide, sodium hydroxide (NaOH), acetone, buffered formalin, paraffin, and toluidine blue were purchased from Merck and the Alfa-Aesar Company.

### 2.2. Synthesis of Mn_2_O_3_ and MnCoO, Mn:Co (75:25 wt.%) Nanoparticles

The nanoparticles were synthesized via the chemical reactions as:$${\mathrm{MnCl}}_{2}+2\mathrm{NaOH}\to \mathrm{Mn}{\left(\mathrm{OH}\right)}_{2}+2\mathrm{NaCl}$$
$$\mathrm{Mn}{\left(\mathrm{OH}\right)}_{2}\overset{\Delta }{\to }\mathrm{MnO}\;+{\;H}_{2}O$$
$${\mathrm{CoCl}}_{2}.6H_{2}O+2\mathrm{NaOH}\text{⟶}\mathrm{Co}{\left(\mathrm{OH}\right)}_{2}+2\mathrm{NaCl}+6H_{2}O$$
$$\mathrm{Co}{\left(\mathrm{OH}\right)}_{2}\overset{\Delta }{\to }\;\mathrm{CoO}+{\;H}_{2}O$$

Manganese oxide nanoparticles (Mn_2_O_3_ NPs) were prepared by dissolving 8.90 g manganese chloride in 70 mL double distilled water using a magnetic stirrer for 2 h. Then, 70 mL NaOH solution (5.60 g/70 mL water) was carefully added drop by drop to the aqueous solution until a homogeneous precipitate powder was formed at pH reaction equal to 10. The resulting powder was filtered, washed, and dried in a muffle furnace at 70 °C overnight. To obtain Mn_2_O_3_ nanostructure, the powder sample was annealed at 500 °C for 2 h in air. Co-doped Mn_2_O_3_ (Mn:Co 75:25 wt.%) was synthesized by dissolving 6.95 g MnCl_2_ in 45 mL double distilled water and 3.45 g CoCl_2_.6H_2_O in 30 mL. The aqueous solutions were added together with continuous stirring for 3 h without heating. After that, 70 mL NaOH was added dropwise to the mixture at constant stirring speed (800 rpm) until a homogeneous solution formed at pH~9. The resulting powder was separated using filter paper then washed with deionized water. Thereafter, it was transferred to the muffle furnace for drying and eventually annealed at 500 °C in air for 2 h.

### 2.3. Characterization of the Prepared Samples

The crystal structure of un-doped and Co-doped Mn_2_O_3_ was identified by X-ray diffraction (XRD, Bruker D8 Discovery diffractometer, Bruker, Billerica, MA, USA) at wavelength (λ)=1.54 Å (CuKα radiation) through 2θ ranging from 15 to

70°. The molecular interaction was recorded by Raman spectroscopy (Horiba LabRAM HR, Horiba, Kyoto, Japan). Energy dispersive X-ray analysis (EDAX; Helios Nanolab. 400, TSS Microscopy, Hillsboro, OR, USA) was employed to determine the elemental composition and purity of the prepared nanopowders. Morphological features, topological architectures and particle size distribution were visualized using scanning electron microscopy (SEM; Helios Nanolab. 400) and transmission electron microscopy (TEM; Hitachi-H-7500, Hitachi, Tokyo, Japan). The influence of thermal treatment on Mn_2_O_3_ and MnCoO stability was examined by Thermogravimetric (TGA, Netzsch Jupiter, Netzsch, Hanau, Germany). The optical absorption and energy gap of the synthesized thin films were estimated from the optical absorbance measurements carried out at room temperature using UV-Vis spectrophotometer Jasco (V-570) (Jasco, Tokyo, Japan).

### 2.4. Mosquito Rearing 

*C. pipiens* larvae were obtained from the Medical Institution of Entomology, Dokie, Cairo, Egypt, and grown in the insect lab at temperature 27 ± 2 °C, relative humidity (RH)70 ± 0%, and 10 h:14 h photoperiod. The mosquito adults were reared in cages (45 × 45 × 45 cm) and fed with a 10% glucose solution. The females lay eggs after feeding on a blood meal. Egg rafts were gathered and put in plastic pots full of water for hatching. Larvae were given fish food and grown in white dishes having distilled water [23]. 

### 2.5. Larvicidal Activity

The synthesized Mn_2_O_3_ and MnCoO NPs were individually tested against the third stage of *C. pipiens* larvae and pupae. The nanostructures were evaluated at different amounts (100, 150, 200, 250, and 300 ppm) [24] with a little alteration. For each test, larvae were put into five replicates. Twenty larvae per replicate were placed into 200 mL sterile water in a plastic beaker covered with mosquito netting and the exact steps were for pupae. Larval and pupal mortality was noticed daily for 72 h. The mortality data were analyzed with one-way analysis of variance (ANOVA) and the lethal concentrations (LC_50_ and LC_90_) at 95% confidence limits were revealed by probit analysis.

### 2.6. Scanning Electron Microscopy (SEM) 

The treated pupae were immersed for 2 h in 2.5% glutaraldehyde for structure fixation and rinsed with 0.1 M cacodylate buffer (pH 7.2) for about 15 min. After that, specimens were put in osmium tetroxide (OsO_4_) for post-fixation for 2 h. Then, they were washed with a buffer, dehydrated in diluted ethanol, embedded in an acetone solution, and, finally, coated with iridium vapor utilizing an imaging sputter coater (Model, EMS 150T ES, Quorum, Kent, UK). The samples were scanned by SEM (Helios Nano-Lab. 400).

### 2.7. Histopathological Examination

The infected larvae samples were gathered from tested groups, fixed in neutral buffered formalin 10%, washed, dehydrated, filtrated, and immersed in paraffin. The paraffin embedded blocks were sectioned at Leica EM KMR2 (Leica, Wetzlar, Germany) ultra-microtome and stained toluidine blue according to [25] with some modifications for histological investigation. Stained sections were investigated by a light microscope (CXL Binocular compound microscope optic, Labomed, Inc., Los Angeles, CA, USA).

## 3. Results and Discussion

### 3.1. Phase Content and Purity of the Chemical Composition

The structure, molecular interaction, and elemental composition of the nanostructured materials were analyzed by XRD, Raman, and EDX spectra. The XRD pattern of the prepared samples calcined at 500 °C was depicted in Figure 1a. As shown, the pattern of the well-defined crystalline structure with diffraction peaks at 2θ= 23.22°, 32.99°, 38.35°, 45.32°, 49.50°, 55.40°, 60.65°, 64.29°, 65.95°, and 67.63°, corresponding to the reflection planes (211), (222), (400), (332), (431), (440), (611), (541), (622), and (631), respectively, ascribed to the cubic phase of pure Mn_2_O_3_ (PDF #97-000-9091) [26]. No further phases related to impurities or unreacted salts were detected [27]. On the other hand, the MnCoO nanocomposite displays a binary phase in which the cobalt oxide (Co_3_O_4_) phase was observed with the reflection planes (111), (220), (311), (222), (331), and (511), according to (PDF No. 43-1003) [28]. As can be seen, the peak intensity of Mn_2_O_3_ was reduced and shifted to lower 2θ by Co^3+^, which supports the structure deterioration and crystalline size reduction. It is well known that many factors effect the host nanomaterial framework, leading to modifications to its microstructure such as ionic radius, concentration ratio, electronegativity, and type of dopant ions. Here, the high Co^3+^ amount inside the Mn_2_O_3_ lattice resulted in phase segregation in addition to the disappearance of some peaks. The impact of cobalt ions (Co^3+^) concentration on the crystalline size $\left(D\right)$ and strain $\left(\varepsilon \right)\;$of the Mn_2_O_3_ lattice was described from Debye’s Scherrer equation expressed as [29]:(1)$$D=\frac{K\lambda }{\beta \;\cos\theta }$$
(2)$$\varepsilon =\frac{\beta \cos\theta }{4}$$
where k is the shape factor given by the constant value 0.94, λ is the wavelength of incident X-ray, β is the full width at half maximum (FWHM), and θ is the reflection angle. The values of the crystallite size were 43.08 nm and 39.12 nm for Mn_2_O_3_ and MnCoO nanocomposite, respectively. Additionally, the strain values were 8.40 $\times 10^{-4}$ and 9.25 $\times 10^{-4}$ for un-doped and Co^3+^ doped Mn_2_O_3_, respectively. The increase in strain was attributed to the crystal defect resulting from cobalt ions substitutions [28,29]. Further, the phase content and lattice defect were defined by Raman spectra through the spectral range from 200 to 3000 cm^−1^. Figure 1b illustrates Mn_2_O_3_ of main characteristic mode at 630 cm^−1^ related to Mn−O vibrations [30], whereas MnCoO of a sharp Raman active mode at 680 cm^−1^ was associated with Co−O−Mn stretching vibration. 

The shift to higher wave number was assigned to the structure distortion and presence of defects. This behavior refers to the generation of strain and more energy states, revealing the strong molecular interaction between Co^3+^ and Mn_2_O_3_. The weak Raman peak observed at 400 cm^−1^ was indexed to the second vibration mode of Mn-O vibrations [31]. The purity and main elemental composition of the nanoparticles were examined from EDAX spectra. As described in Figure 2a, the Mn_2_O_3_ spectrum composed of Mn and O elements of weight percent 74.22 and 25.78 wt.%, respectively. In Figure 2b, the EDAX spectrum proved that the cobalt atoms were successfully incorporated into the host Mn_2_O_3_ (17.12 wt.%), resulting in the production of MnCoO nanocomposite [32,33].

### 3.2. Morphological and Thermal Analysis 

The morphological results obtained from the SEM images are presented in Figure 3a,b. As demonstrated in Figure 3a, the particles of Mn_2_O_3_ have a cubic shape linked with nanoneedles. Further, the surface appears irregular and rough owing to the particles’ aggregation. Figure 3b describes the surface of MnCoO in flake-like structures, and the particles appear uniform with high density [34,35]. To visualize the mean size and particles distribution, TEM spectroscopy was performed. The TEM image has proven the cubic shape of Mn_2_O_3_ with average particle size ~50 nm (Figure 3c). Moreover, Figure 3d describes the particles of MnCoO in nanoflakes with a large surface area [35,36,37]. The nanoflake structure has significant importance in biotechnological applications, such as for anticancer, antimicrobial, and insecticidal purposes. 

The impact of annealing temperature on the nanoparticles’ behavior was studied from TGA measurements in a nitrogen atmosphere between 50 and

800 °C, as described in Figure 4. The TGA curve of Mn_2_O_3_ shows a weight loss of 7% at heat treatment between 50 and 170 °C, attributed to the decomposition of water molecules. Through the thermal annealing from 260 to 280 °C, the weight loss reaches 10% related to the removal of hydroxide molecules. The final weight loss of 2% between the calcination temperature 400 to 500 °C was associated with the formation of pure Mn_2_O_3_ [38,39]. The nanoflakes show similar behavior, with higher weight loss between 50 and 400 °C reaching 18% at 500 °C, attributed to the decomposition of water, salt, or hydroxide molecules. 

### 3.3. Optical Analysis 

The absorbance spectra of un-doped and Co-doped Mn_2_O_3_ coated on glass substrates were measured at wavelength$\;\left(\lambda \right),$ changed from 350 to 800 nm. As illustrated in Figure 5a, the thin films have optical absorbance peak at approximately 395 within the visible region. The absorption coefficient (α) as a function of photon energy (hν) is depicted in Figure 5b. MnCoO exhibits high optical absorption due to the strong photo-electron interaction [40]. The red shift to higher energy is owing to Co^3+^ dopants inside the Mn_2_O_3_ framework and the generation of new energy levels within the energy gap. Furthermore, the optical band gap was evaluated from Tauc’s relation, expressed by the following equations [41]:(3)$$\left(\alpha h\upsilon \right)=K{\left(h\nu -E_{g}\right)}^{1/2}$$
(4)$$\alpha =\frac{2.303\;A}{t}$$
where A is the optical absorbance, hν is the energy of incident photons, K is a constant, t is the film thickness, and α is the absorption coefficient. From $\;{\left(\alpha h\nu \right)}^{2}$ versus (hν) plots in Figure 5c, the energy gap ($E_{g})$ was evaluated by extending the linear portion of ${\left(\alpha h\nu \right)}^{2}$ to ${\left(\alpha h\nu \right)}^{2}=0$. The results confirmed that the thin films of direct band gap in which the obtained curves are linear for n = 1/2. The values of the energy gap for Mn_2_O_3_ and Co-doped Mn_2_O_3_ were 2.95 eV and 2.80 eV, respectively, which prove the increase in free charge carriers by absorbing photon energy and the transfer of more electrons to the conduction band [42,43]. 

### 3.4. Toxicity of Nanoparticles to Larvae and Pupae of C. Pipiens

The synthesized Mn_2_O_3_ NPs and MnCoO nanocomposites showed a dose-dependent insecticidal impact against mosquito *C. pipiens* larvae and pupae. The MnCoO nanocomposites showed an enhanced larvicidal effect, causing 100% larval mortality compared to Mn_2_O_3_ NPs which recorded a mortality rate of 88% at a high concentration (Table 1). Moreover, compared to Mn_2_O_3_ nanoparticles, the MnCoO nanocomposites observed higher lethality against the third larval instars of *C. pipiens*, with LC_50_ and LC_90_ values of 159.78 and 263.20 ppm, respectively. The results of the pupicidal activity of fabricated Mn_2_O_3_ NPs and MnCoO NPs against the pupa of *C. pipiens* are presented in Table 1. Considerable mortality was obvious after the treatment at high doses; the pupa showed restless motion for a period, with unusual wagging, and finally died. The percentage of pupal mortality was directly proportional to concentrations. A high efficacy was observed in applying of MnCoO as pupicidal against the pupa with LC_50_ = 227.40 ppm, compared to Mn_2_O_3_ (LC_50_ = 277.83 ppm). These findings are consistent with Mohamed et al. [21], who recorded that the Co_3_O_4_ NPs caused 100% larval mortality of *C. pipiens* at the concentration of 600 ppm, and 30% at 100 ppm, and the value of LC_50_ was 250.45 ppm. Furthermore, Kainat et al. [44] detected that a Co_3_O_4_ nanostructure has a larvicidal activity against *Aedes aegypti* with a mortality of 67.2% at 400 ppm, compared to MgO nanoparticles of 49.2% death rate at the same dose. Additionally, the modified MnCoO nanoflakes have a high surface area, causing a disorder of *C. pipiens* enzymatic activity, leading to the death of immature stages according to [12]. The growth inhibition and death of mosquito immature stages which were exposed to metal oxide nanostructure and nanocomposites are attributed to the release of reactive oxygen species (ROS) from metal oxide dissociation when they come in contact with larval body cells [45,46]. The toxicity of metal oxides to mosquito larvae starts when they enter the mosquito cells and cause DNA nicking [47]. Furthermore, metal oxide NPs bind to the cell membrane, leading to cell membrane destruction [48]. Additionally, an increase in metal ions disrupts the cellular equilibrium and results in cellular stress, which in turn leads to cellular leakage and mosquito cell death [49,50]. Thus, the unique characteristics of metal oxide nanostructures create opportunities to utilize them as potential mosquitocidal agents. Herein, the MnCoO nanoflakes have revealed better performance against larvae compared to Mn_2_O_3_ thanks to their excellent structural-morphological properties.

### 3.5. Histological Observations of Treated Larvae and Pupa 

The histological study of the third larval instars of *C. pipiens* exposed to a median lethal concentration of Mn_2_O_3_ and MnCoO NPs revealed observable effects. Before treatment, in the midgut of the control larvae, a regular epithelial layer, normal columnar epithelial cells (CE) with nuclei (N), and a peritrophic membrane (PM) were observed to be complete (Figure 6a), and the brush border of the midgut was perfectly tight and fit (Figure 6b,c). This coincided with Abutaha et al. and Al-Mekhlaf [51,52], who illustrated the normal anatomical structure of the middle intestine of *C. pipiens* larvae. Moreover, the structure of the mid-gut epithelium has a considerable role in all stages of the cycle of insect life [53]. Additionally, the peritrophic membrane (PM), a semi-permeable lining of the mid-gut, protects the underlying epithelium against food abrasion and microbial attacks in addition to its role in facilitating the digestive process [54]. On the other hand, after treatment the damages caused in the mosquito’s middle intestine have been highlighted in the affected *C. pipiens* larvae. The histopathology of Mn_2_O_3_-treated larvae showed disordered and ruptured epithelial cell layers, in addition to the disappearance of the peritrophic membrane (PM) and the microvilli of the brush border (MV). Moreover, severely affected gut content (GC), circular muscle (CM), and adipose tissue (AT) were noted. In addition, the appearance of nanoparticles inside the gut lumen and under the cuticular structure was elucidated in (Figure 7a–d), while the treated larval midgut with MnCoO nanostructure in (Figure 8a–d) illustrated a complete destruction of mid-gut epithelial cells and resulted in the widening of intercellular spaces, cytoplasmic vacuolization, and the disappearance of the peritrophic membrane and nuclei. Nanosheets of MnCoO have been seen in the gut lumen and they affect fat cells and muscle fibers. In this histological study, the midgut showed more destruction in the gut epithelial cells, muscles, and adipose tissue. Hence, the observation evidenced the clear deposition of NPs in the midgut region and their lumen. The deformations in this study agreed with Sundararajan et al. [55], who reported that the synthesized Au NPs cause a high mortality rate in *A. aegypti* immature stages and result in histological alterations in the midgut columnar epithelium cells, digestive tract, and cuticle. The mode of action of metal oxide nanoparticles in the larval middle intestine is due to its impact on digestive tract enzymes, structural deformation in DNA, and the generation of reactive oxygen species, which cause the denaturation of enzymes and affecting the functions of organelles [56,57]. Fiaz et al. and Martínez et al. related the digestive cell death of the treated midgut cells to cytoplasm vacuolization followed by cell damage and the release of cell debris into the gut lumen [58,59,60].

Further, different pupal abnormalities are illustrated in treated pupa examined by SEM. The deformed pupa has different colorations, which convert from dark yellowish to blackish discoloration. Additionally, the treated pupa of *C. pipiens* with Mn_2_O_3_ showed different malformations and accumulations of nanostructure in the puparium, particularly in mesothoracic (MW) and metathoracic (MtW) wings (Figure 9a–d). For MnCoO treatment, the malformation was observed in the abdominal segment and especially in the intersegmental membrane (IM), genital lobe (Gl), paddle (Pa), and midrib (Mr), as they showed a high aggregation of nanoparticles (Figure 10a–e) which may be the reason for pupal death [61]. The nanostructure compounds directly impact epidermal cells responsible for producing enzymes and the cuticular oxidation process. With respect to the larger surface and smaller size of the NPs, they enable them to enter the larval body and disrupt their normal life cycle by interrupting cell division and breathing [62]. Nalini et al. [63] reported that Ag NPs synthesized by the *Artemisia nilagirica* leaf showed a significant delay in pupation and an abnormal development of the wing and larval body after Ag NPs treatment.

## 4. Conclusions

Mn_2_O_3_ and MnCoO nanostructures were successfully prepared by a cost-effective co-precipitation route and analyzed by XRD, Raman, EDX, SEM, and TEM micrographs. The microstructure analysis revealed a modification of the Mn_2_O_3_ topological nature by Co^3+^ (0.25 wt.%) support. The 2D nanoflakes, MnCoO, exhibited a large surface-to-volume area, which provides a strong interaction with the larvae body using tiny amounts of the nanoparticles. TGA analysis demonstrated the formation of Mn_2_O_3_ and MnCoO at annealing temperatures between 400 and 500 °C, with high thermal stability. The energy gap $E_{g\;}$was calculated within the visible region less than 3 eV. The larvicidal and pupicidal activity of the prepared nanoparticles were examined at different concentrations. In addition, the nanoparticles induce a pathological alteration in the larval midgut structure, fat tissue, and pupal morphological deformation. The obtained results confirmed that the superior structural-morphological features enable MnCoO nanoflakes to be used as a mosquitocidal agent with high efficiency, based on its large surface area and highly reactive oxygen species. 

## Figures and Tables

**Figure 1 micromachines-14-00567-f001:**
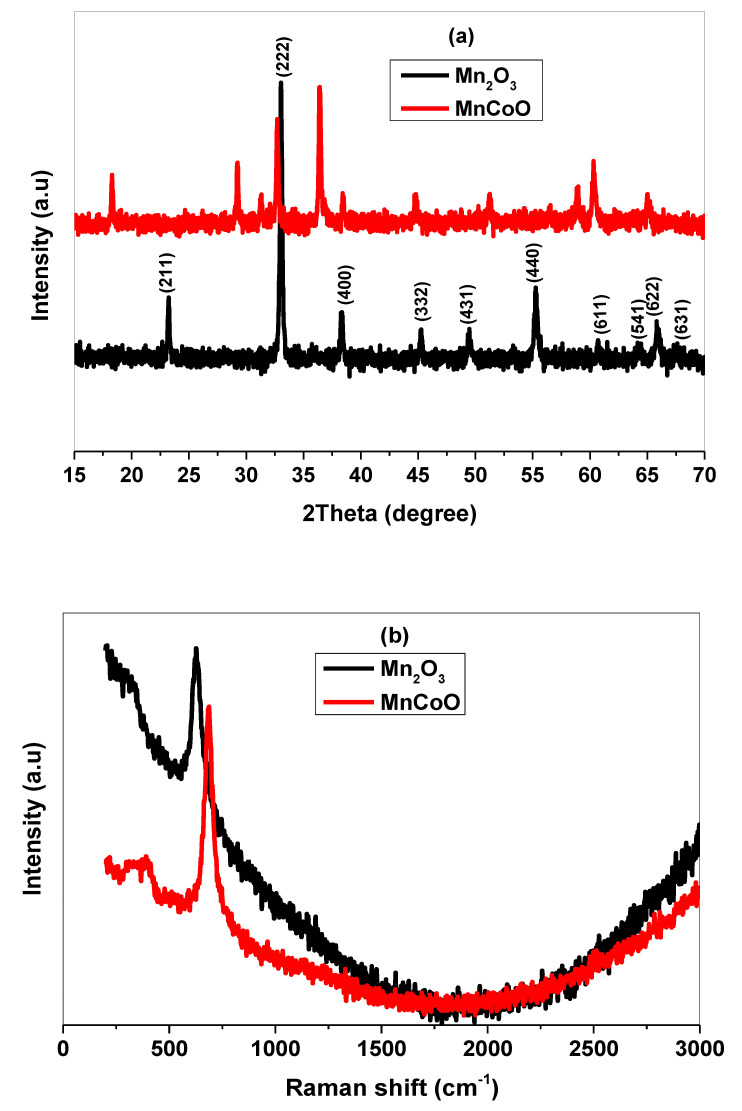
(**a**) XRD pattern, (**b**) Raman spectra of Mn_2_O_3_, and MnCoO nanocomposite.

**Figure 2 micromachines-14-00567-f002:**
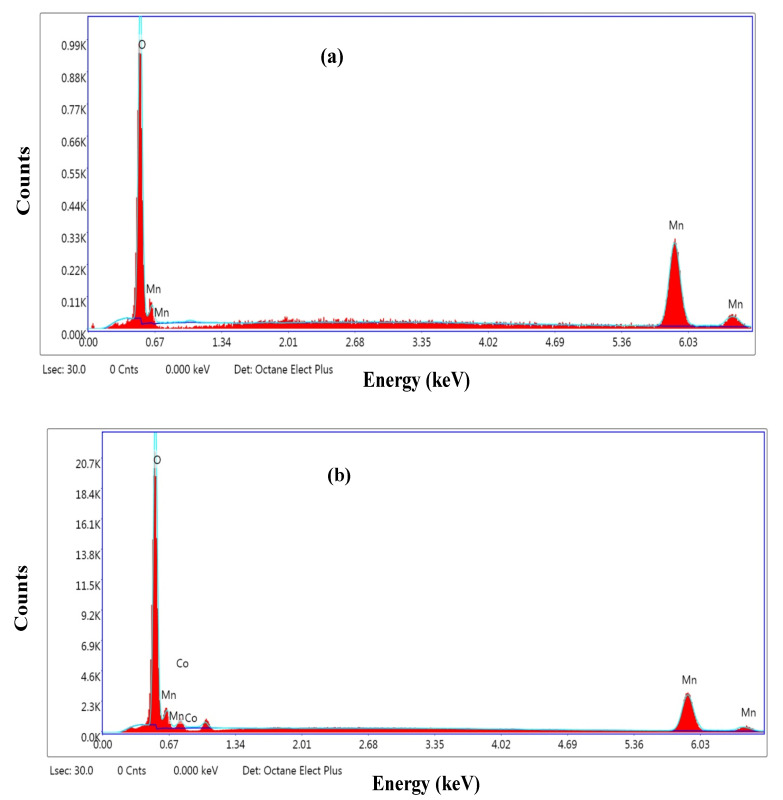
EDAX spectra of (**a**) Mn2O3 and (**b**) MnCoO nanocomposite.

**Figure 3 micromachines-14-00567-f003:**
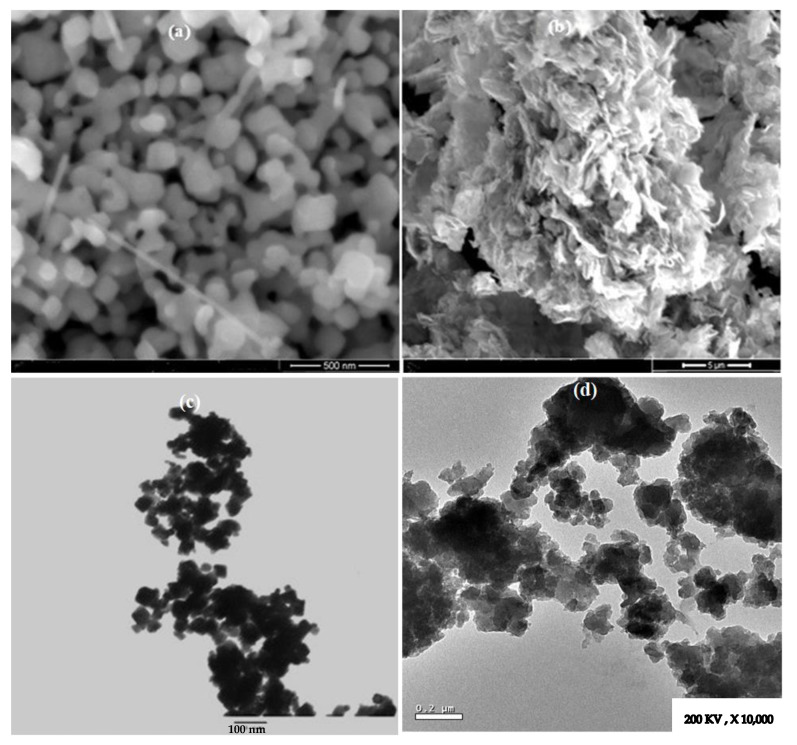
(**a**,**b**) SEM micrographs, (**c**,**d**) TEM micrographs of Mn_2_O_3_ and MnCoO nanoflakes.

**Figure 4 micromachines-14-00567-f004:**
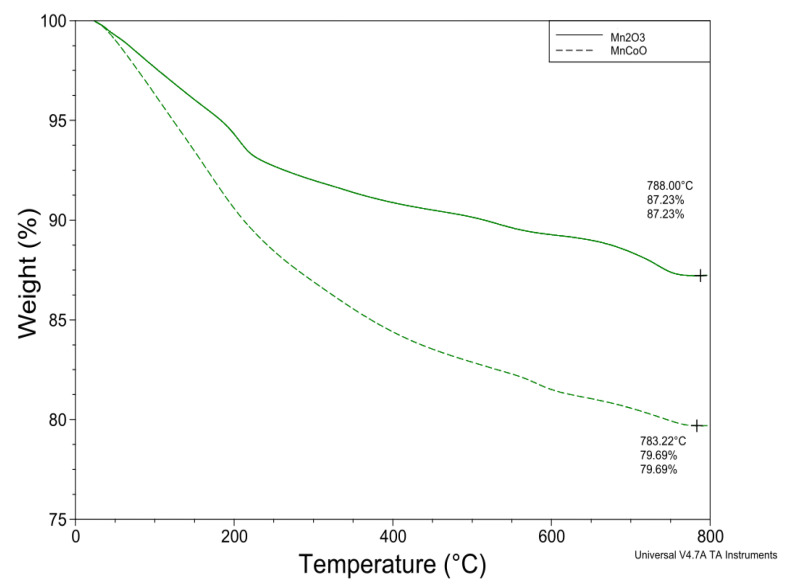
TGA graph of un-doped and Co-doped Mn_2_O_3_.

**Figure 5 micromachines-14-00567-f005:**
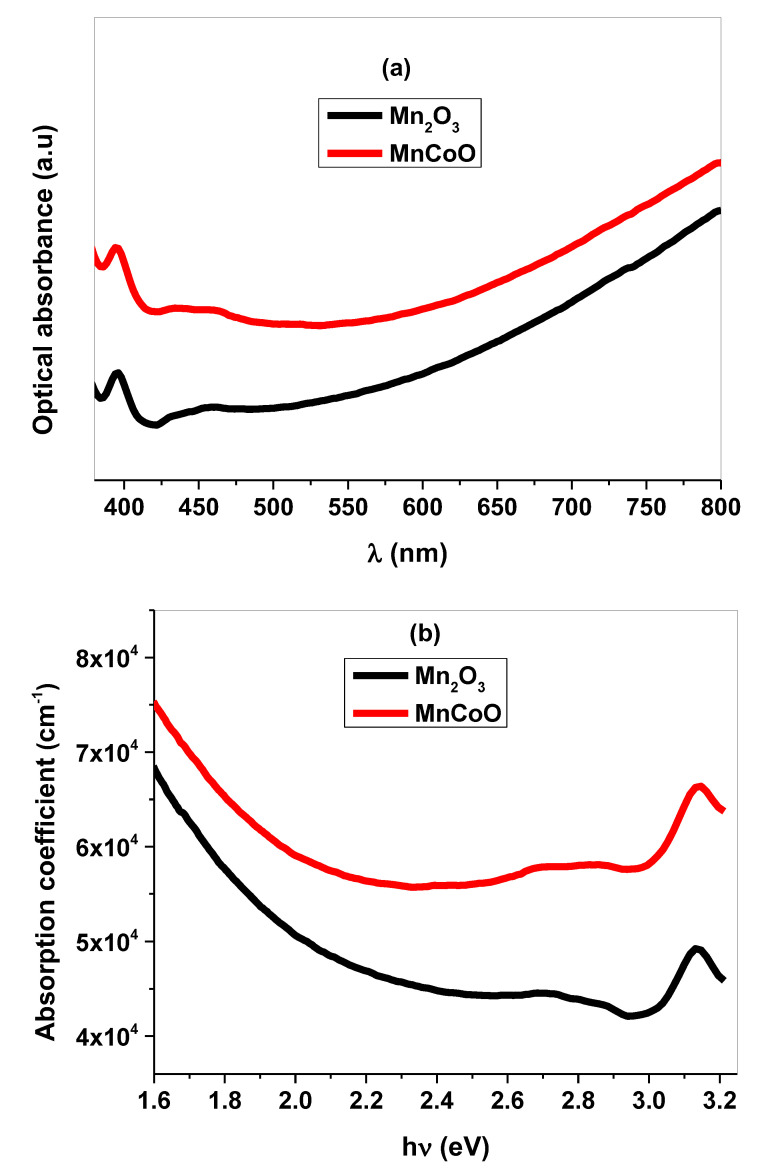
(**a**) Optical absorbance, (**b**) absorption coefficient, and (**c**) optical band gap plot of the fabricated thin films.

**Figure 6 micromachines-14-00567-f006:**
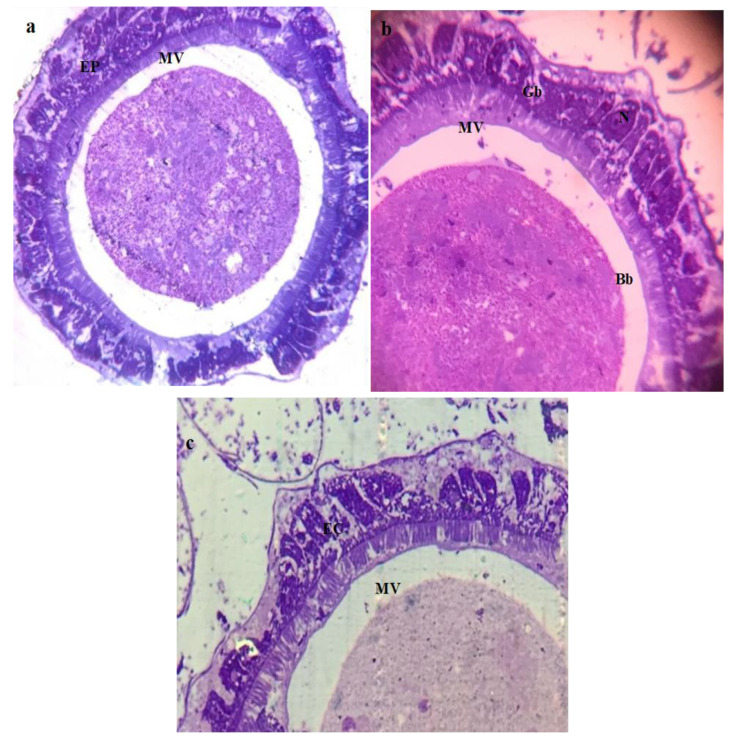
Photomicrographs of a cross-section of control larva of *C. pipiens*: (**a**) normal midgut with columnar epithelial cells (EC), and (**b**,**c**) the cells have nuclei (N), Microvilli (MV), brush border (Bb), and goblet cells (Gb) among epithelial cells (EP).

**Figure 7 micromachines-14-00567-f007:**
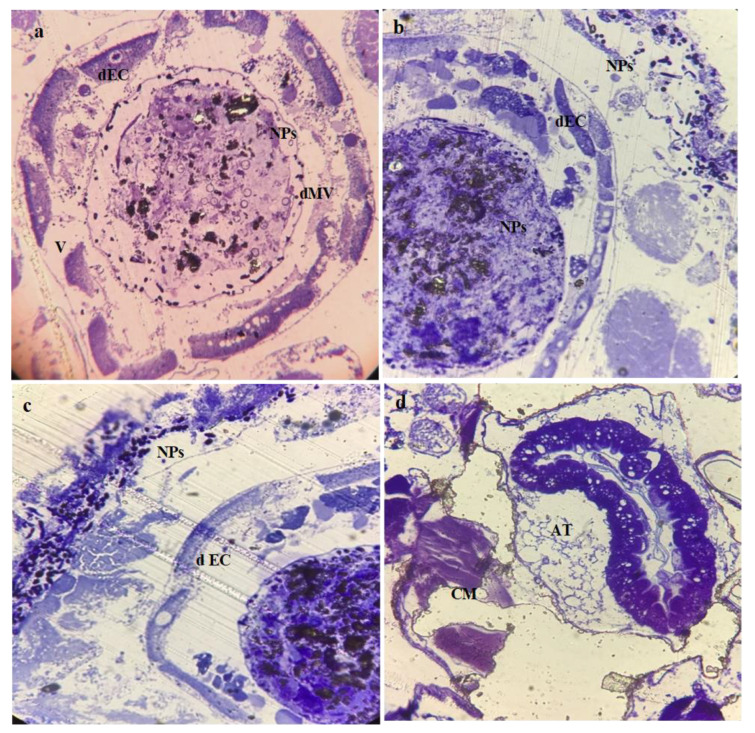
Photomicrographs of a cross-section of *C. pipiens* larvae treated with Mn_2_O_3_ NPs: (**a**,**b**) treated midgut with damaged columnar epithelial cells (dEC) and vacuoles (V), destructive Microvilli (dMV), and (**c**,**d**) destructive epithelial cells (d EC), adipose tissue (AT), rupture circular muscles (CM), and nanoparticles (NPs).

**Figure 8 micromachines-14-00567-f008:**
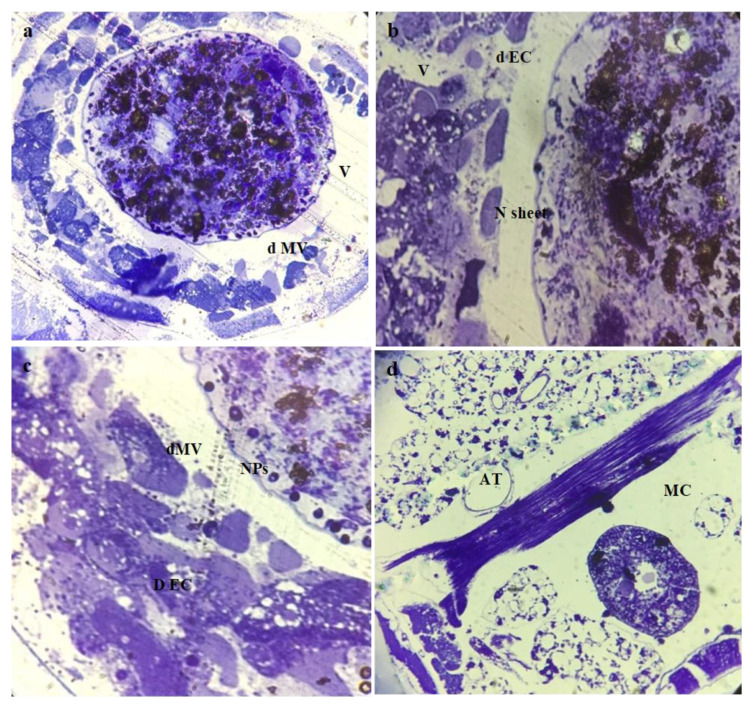
Photomicrographs of a cross-section of *C. pipiens* larvae treated MnCoO nanocomposites: (**a**,**b**) midgut with damaged columnal epithelial cells (dEC) with vacuoles (V), destructive Microvilli (dMV) and accumulation of nanosheet (N sheet), (**c**) appearance of nanoparticles (NPs) in gut lumen and vacuolated cell, and (**d**) adipose tissue (AT) and circular muscles (CM).

**Figure 9 micromachines-14-00567-f009:**
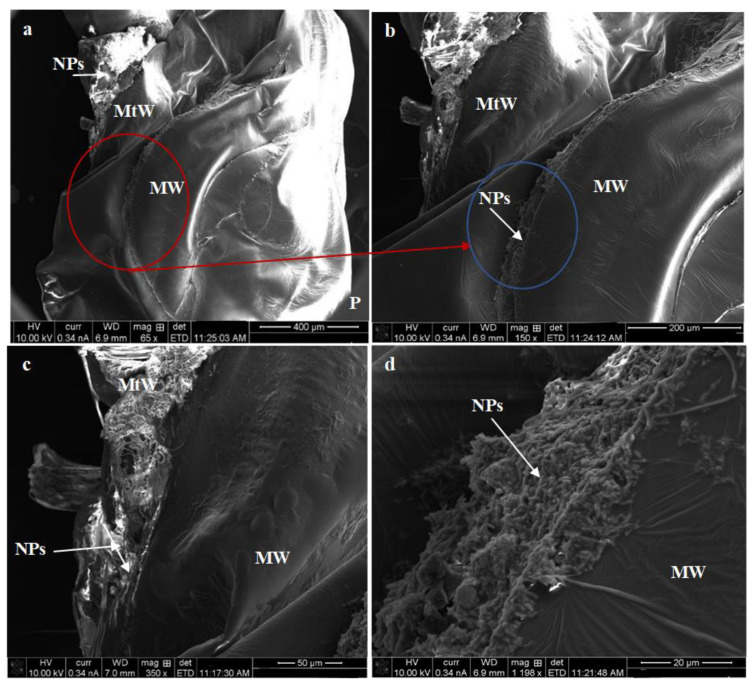
SEM images of *C. pipiens* pupal stages treated with Mn_2_O_3_ NPs: (**a**,**b**) deformed puparium in mesothoracic (MW) and metathoracic (MtW) wing, and (**c**,**d**) accumulation of nanoparticles in thoracic part (MtW: metathoracic wing, MW: mesothoracic wing, NPs: nanoparticles).

**Figure 10 micromachines-14-00567-f010:**
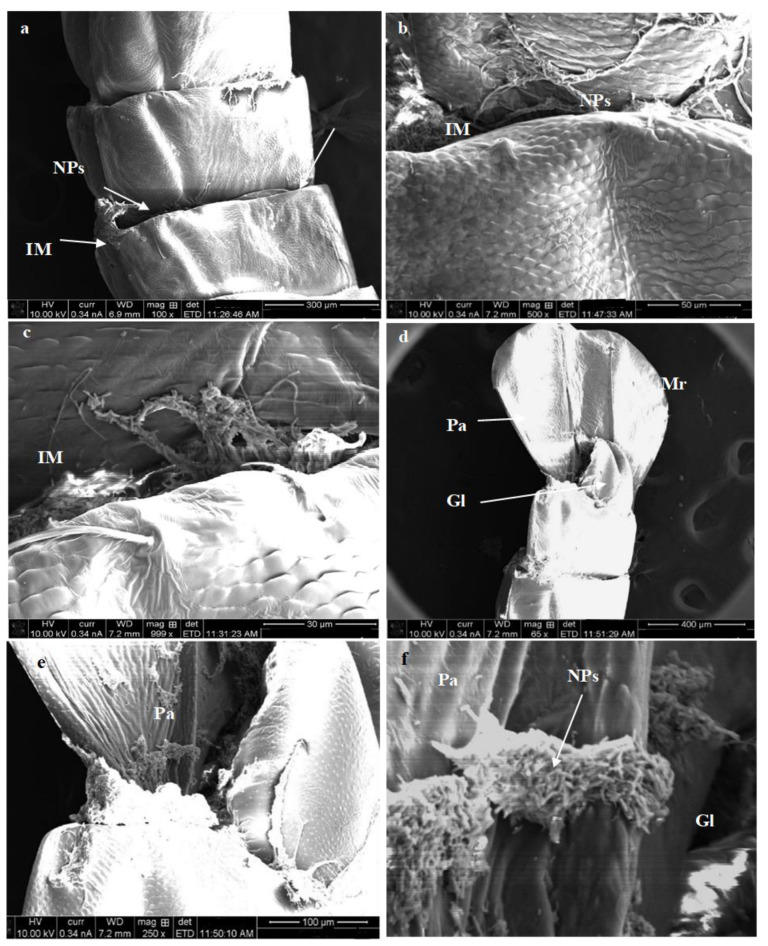
SEM images of the abdominal segments and paddle region of *C. pipiens* pupal stages treated with CoMnO nanocomposites: (**a**–**c**) malformation in the abdominal segment, especially in the intersegmental membrane and aggregation of NPs, and (**d**–**f**) high aggregation of nanoparticles in genital lobe, paddle, and midrib (GI: genital lobe, IM; intersegmental membrane, Mr: midrib, NPs: nanoparticles, Pa: paddle).

**Table 1 micromachines-14-00567-t001:** Dose-dependent larvicidal and pupicidal activity against *C. pipiens* species.

Nanoparticles		Larval Mortality	LC_50_	LC_90_ (ppm)	Regression Equation	Pupal Mortality	LC_50_ (ppm)	Regression Equation
Conc.(ppm)		(Mean ± S.E)	(95%LCU: UCL)		(X^2^)	(Mean ± S.E)	(95%LCU: UCL)	(X^2^)
Mn_2_O_3_	100	25 ± 0.478 ^a^	182.94 (168.90:196.0)	324.56 (300.11: 359.40)	Y = −2.25 + (X × 0.125) (9.87)	10 ± 0.288 ^a^	277.83 (258.14:305.64)	Y= −2.5 + 0.1 X (8.24)
	150	36 ± 0.408 ^b^				15 ± 0.478 ^b^		
	200	56 ± 0.577 ^c^				30 ± 0.645 ^c^		
	250	70 ± 0.645 ^d^				45 ± 0.629 ^d^		
	300	88 ± 1.22 ^e^				60 ± 0.912 ^d^		
CoMnO	100	36 ± 0.408 ^a^	159.78 (147.10:171.20)	263.20 (246.26:286.13)	Y = −1.83 + (X × 0.133) (14.06)	16 ± 0.408 ^a^	227.40 (212.53:244.44)	Y= −2.75 + 0.125 X (6.46)
	150	48 ± 0.324 ^b^				16 ± 0.408 ^a^		
	200	63 ± 0.478 ^c^				42 ± 0.645 ^c^		
	250	86 ± 0.645 ^d^				58 ± 0.288 ^d^		
	300	100 ± 0.00 ^e^				71 ± 1.030 ^d^		

LC_50_ and LC_90_ lethal concentration that kills 50 and 90% of the exposed larvae, respectively; UCL, upper confidence limit; LCL, lower confidence limit; X^2^, chi-square; df, degrees of freedom, Significant at *p* ≤ 0.05; SE: stander error. Letters indicate degree of significant based on Tukey’s HSD tests between concentrations.

## Data Availability

Data generated or analyzed during this study are included in this published article.

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
