# Peer review of "Modulation of the Morphological Architecture of Mn2O3 Nanoparticles to MnCoO Nanoflakes by Loading Co3+ Via a Co-Precipitation Approach for Mosquitocidal Development"

_micromachines, 2023, doi:10.3390/mi14030567_

Round 1

Reviewer 1 Report

The submitted article presents synthesized Mn2O3 nanoparticles and MnCoO nanocomposite as mosquitocidal agents. The nanoparticles were prepared by by cost-effective co-precipitation route. The performed TGA analysis proved their high thermal stability. The energy gap (Eg) of Mn2O3 was measured within the visible spectrum of the value 2.92 25 eV which reduced to 2.72 eV with doping support. The obtained nanoparticles induce pathological alteration in larval midgut structure, fat tissue, and pupal morphological deformation. Therefore, the created materials could be used as mosquitocidal agent with high efficiency based on highly reactive oxygen species.

Specific points requiring attention are detailed below.

Comment 1. Include citations in the Introduction section after sentences:

Nanomaterials and their derivatives including nobel nanometals (i.e Au, Ag, Ru), carbonate nanomaterials (i.e CNTs, GO), and na-50 nometal oxides have gained great attention in the present decade.

Nowadays, metal oxide 51 semiconductors are used in electronics, solar cells, and photocatalysis. ZnO, CuO, MgO, and AgO nanoparticles (NPs) have approved excellent performance in biomedical, envi-53 ronmental remediation, food security, and agriculture.

Comment 2. Remove the years after the authors names in the Introduction section:

M. Kelani et al., (2022)

Jiansheng Lin et al., (2022)

Comment 3. Specify the producer of the chemicals in the Section 2.1. Reagents.

Comments 4. Improve the quality of the chemical reaction in page 3.

Comments 5. Use the template of the Journal. Include the Figures into the text.

Comment 6. Use the template of the Journal for the References.

Comment 7. Rearrange the Table 1 and 2.

Comment 8. Emphasize on the novelty of the paper. 

Author Response

Changes/Modifications in Response to Editor and Reviewers Comments

Manuscript ID: micromachines-2198622

Editors-in-Chief

Thank you very much for your e-mail concerning the revision of our manuscript entitled: Modulation of the morphological architecture of Mn2O3 nanoparticles to MnCoO nanoflakes by loading Co3+ via co-precipitation approach for mosquitocidal development

I am grateful for handling my manuscript and for reviewers’ diligent efforts for improving my manuscript. The reviewers’ suggestions have been implemented and the manuscript has been modified accordingly. I have made a point-by-point response to each comment raised by the reviewers as following:

Note: The English language editing has been done

Reviewer's comments to authors:

Reviewer #1:

Comment 1.

Include citations in the Introduction section after sentences: Nanomaterials and their derivatives including nobel nanometals (i.e Au, Ag, Ru), carbonate nanomaterials (i.e CNTs, GO), and nanometal oxides have gained great attention in the present decade. Nowadays, metal oxide semiconductors are used in electronics, solar cells, and photocatalysis. ZnO, CuO, MgO, and AgO nanoparticles (NPs) have approved excellent performance in biomedical, environmental remediation, food security, and agriculture.

Reply: Thanks a lot for your valuable recommendations. The citation references [10,11,12,13] were added after the mentioned sentences in the introduction and highlighted in the revised manuscript. Thank you so much.

The references

  1. Cestaro, R.; Schweizer, P.; Philippe, L.; Maeder, X.; Serr, A. Phase and microstructure control of electrodeposited Manganese Oxide with enhanced optical properties. Appl. Surface Sci. 2022, 580, 152289.
  2. Noelson, E.A.; Anandkumar, M.; Marikkannan, M.;  Ragavendran, V.; Sagadevan, T.A.; Annaraj, S.J.; Mayandi, J. Excellent photocatalytic activity of Ag2O loaded ZnO/NiO nanocomposites in sun-light and their biological applications. Chemical Physics Letters. 2022, 796, 139566.
  3. Mohamed, R.A.; Ghazali, N.M.; Kassem, L.M.; Elgazzar, E.; Mostafa, W.A. Synthesis of MnCoO/CNT nanoflakes for the photocatalytic degradation of methyl orange dye and the evaluation of their activity against Culex pipiens larvae in the purification of fresh water. RSC Adv. 2022, 12. 29048.
  4. Nethravathi , P.V.; Suresh, D. Silver-doped ZnO embedded reduced graphene oxide hybrid nanostructured composites for superior photocatalytic hydrogen generation, dye degradation, nitrite sensing and antioxidant activities. Inorganic Chemistry Communications 2021, 134, 109051.

Comment 2.

Remove the years after the author’s names in the Introduction section: M. Kelani et al., (2022), Jiansheng Lin et al., (2022)

Reply: The years were removed after the mentioned authors in the introduction section and the modifications were added and highlighted in the revised manuscript. Thanks a lot for your suggestions.

Comment 3.

Specify the producer of the chemicals in the Section 2.1. Reagents.

Reply: Done, the producer companies of the chemicals were added to the experimental section and the paragraph was highlighted and adderd in the revised manuscript according to your suggestions. Thanks a lot

Comment 4.

Improve the quality of the chemical reaction on page 3.

Reply: Done, the quality of the chemical reaction was improved and highlighted in the revised manuscript. Thanks a lot for your revision of our manuscript.

Comment 5.

Use the template of the Journal. Include the Figures in the text.

Reply: Done, all the figures were modified in the revised manuscript according to your valuable recommendations. Thank you so much.

Comment 6.

Use the template of the Journal for the References.

Reply: Done, all references were rewritten according to the journal template. Thank you so much.

Comment 7.

Rearrange the Table 1 and 2.

Thank you so much. Done, the two tables were merged in table 1 and the data was rearranged according to your suggestions.

Comment 8.

Emphasize the novelty of the paper.

Reply: The impact of the synthesized nanosheets on the improvement the larvicidal activity was mentioned in the introduction and results and discussion sections in the revised manuscript. Thanks a lot for your revision of our manuscript. 

Reviewer 2 Report

The paper describes "Modulation of the morphological architecture of Mn2O3 nanoparticles to MnCoO nanoflakes........".

Following are my comments to improve the manuscript;

1: Figure 2 SEM image and TGA need to improve the quality/resolution. Please remove the written content in the SEM image which can be mentioned only scale in the image. Also for the SEM image of Figures 7 and 8.

2: Provide the average size of NPs and zeta potential.

3: All figures need to present scientifically (presentation or very poor).

4: What about the XPS and FTIR analysis of the NPs.

5: Merge Table 1 and Table 2.

Author Response

Changes/Modifications in Response to Editor and Reviewers Comments

Manuscript ID: micromachines-2198622

Editors-in-Chief

Thank you very much for your e-mail concerning the revision of our manuscript entitled: Modulation of the morphological architecture of Mn2O3 nanoparticles to MnCoO nanoflakes by loading Co3+ via co-precipitation approach for mosquitocidal development

I am grateful for handling my manuscript and for the reviewers’ diligent efforts for improving my manuscript. The reviewers’ suggestions have been implemented and the manuscript has been modified accordingly. I have made a point-by-point response to each comment raised by the reviewers as following:

Note: The English language editing has been done

Reviewer's comments to authors:

Reviewer #2:

Comment 1:

Figure 2 SEM image and TGA need to improve the quality/resolution. Please remove the written content in the SEM image which can be mentioned only scale in the image. Also for the SEM image of Figures 7 and 8.

Reply: Done, the quality of SEM images and TGA were improved according to your valuable recommendations. The modified images were added in the revised manuscript. Thank you so much.

Comment 2:

Provide the average size of NPs and zeta potential.

Reply: The crystallite size of the prepared samples was calculated from the Scherrer equation. Also, the lattice strain was calculated. The paragraph was added and highlighted in the revised manuscript. Thanks a lot for your revised our manuscript. 

Comment 3:

All figures need to present scientifically (presentation or very poor).

Reply: Done, all figures were presented scientifically. The figure captions were improved and the figures were renumbered and highlighted in the revised manuscript. Thank you so much for your valuable recommendations and comments.

Comment 4:

What about the XPS and FTIR analysis of the NPs?

Reply: Thank you so much, the microstructure, composition, and phase of the prepared nanomaterials was confirmed by using XRD, SEM, TEM, EDX, and Raman techniques. FTIR and XPS are also important for structure and molecular interaction analysis, we apologize there is not enough time for more measurements.

Comment 5:

Merge Table 1 and Table 2

Thank you so much. The two tables were merged in table 1 and the data was rearranged according to your suggestions.

Round 2

Reviewer 1 Report

I recommend accepting the manuscript in the present form.

Reviewer 2 Report

Congratulations to all authors.